# Non-Coding RNAs and Nucleosome Remodeling Complexes: An Intricate Regulatory Relationship

**DOI:** 10.3390/biology9080213

**Published:** 2020-08-07

**Authors:** Benjamin J. Patty, Sarah J. Hainer

**Affiliations:** Department of Biological Sciences, University of Pittsburgh, Pittsburgh, PA 15260, USA; bjp86@pitt.edu

**Keywords:** chromatin, nucleosome remodeling, non-coding RNA (ncRNA), transcription, lncRNA, eRNA, SWI/SNF, CHD, ISWI, INO80

## Abstract

Eukaryotic genomes are pervasively transcribed, producing both coding and non-coding RNAs (ncRNAs). ncRNAs are diverse and a critical family of biological molecules, yet much remains unknown regarding their functions and mechanisms of regulation. ATP-dependent nucleosome remodeling complexes, in modifying chromatin structure, play an important role in transcriptional regulation. Recent findings show that ncRNAs regulate nucleosome remodeler activities at many levels and that ncRNAs are regulatory targets of nucleosome remodelers. Further, a series of recent screens indicate this network of regulatory interactions is more expansive than previously appreciated. Here, we discuss currently described regulatory interactions between ncRNAs and nucleosome remodelers and contextualize their biological functions.

## 1. Introduction

Many cellular processes occur within the cell nucleus where access to the information encoded with genomic DNA is essential. Eukaryotic genomic DNA is packaged within the nucleus as chromatin– a DNA, RNA, and histone protein structure consisting of repeating elements referred to as nucleosomes [1]. Eukaryotes have evolved several mechanisms to alter chromatin structure, including protein complexes [2], DNA modifications [3], post-translational histone modifications [4], and histone variants [4], which regulate chromatin accessibility for all DNA-templated processes. Chromatin-regulatory proteins can catalyze the addition or removal of DNA modifications or post-translational histone modifications, such as DNA methyltransferases [3] or histone modifying enzymes [5], respectively. Another family of chromatin-regulating protein complexes, ATP-dependent nucleosome remodelers (nucleosome remodelers), utilize ATP hydrolysis to modify nucleosome positioning, composition, and assembly [2,6]. By acting on nucleosomes to alter chromatin structure, nucleosome remodelers permit or restrict access to DNA for protein machinery responsible for DNA repair, DNA replication, and most important for this review, transcription. Due to the important nature of their actions, nucleosome remodelers have emerged as critical regulators of transcription in eukaryotes. 

Recently, it has been shown that at least 75% of the human genome is transcribed at detectable levels [7]. This is striking given that only approximately 1–2% of this genome encodes information for proteins [8]. Traditionally, RNA transcripts generated from non-protein-coding regions of the genome were thought of as transcriptional noise sometimes referred to as “junk”. However, more recent research has shown non-protein-coding, or non-coding RNAs (ncRNAs), are numerous, diverse in form, and serve significant biological function [9,10]. ncRNAs are sorted into classes defined by functionality, length, or from the genomic element from which they originate. Well-known and characterized ncRNA classes include ribosomal RNAs (rRNAs) and transfer RNAs (tRNAs), playing pivotal roles in translation. Research focused on other ncRNA classes, such as long intergenic non-coding RNAs (lincRNAs) and microRNAs (miRNAs), have identified that transcripts from these classes serve important regulatory roles in coding transcription [11,12]. Less well characterized ncRNA classes emerge from functional genomic elements, such as enhancer RNAs (eRNAs), promoter proximal transcripts (PROMPTs), or telomeric repeat containing RNAs (TERRAs).

Largely due to advances in genomics and RNA-protein interaction probing techniques, the identification of interactions between nucleosome remodelers and ncRNAs has emerged as important in regulating one another and therefore function. In several cases, these interactions have been shown to modulate the activity and targeting of nucleosome remodelers [13,14]. Remodeler-based mechanisms to limit non-coding transcription have been observed in eukaryotes previously [7,15,16]. However, these novel interactions call to question if remodeler-based regulation of non-coding transcription could serve as a self-regulatory feedback mechanism. Here, we discuss current literature of regulatory interactions between nucleosome remodelers and ncRNAs and provide perspective on questions that, if addressed, could clarify this interesting and important trend in chromatin biology.

## 2. Nucleosome Remodelers

Nucleosomes are an obstacle preventing the transcriptional machinery access to DNA. Nucleosome remodelers are recruited to chromatin through various mechanisms, where they utilize ATP hydrolysis to alter nucleosome occupancy, positioning, or composition to promote or prevent transcription at sites across the genome. 

Classifications of nucleosome remodelers are primarily driven by the protein domain content of their ATPase subunits. All remodeler ATPases contain a set of ATP-binding/DNA translocation protein domains (the DExx and helicase superfamily c-terminal [HELICc] domains), which classifies them within the RNA/DNA helicase superfamily 2. However, remodeler ATPases are then further classified into four remodeler subfamilies due to the presence of accessory protein domains found within the ATPase subunit [17]. Initially identified in yeast [18,19], the four nucleosome remodeler families are evolutionarily conserved in higher eukaryotes, with a higher diversity of subfamily members likely driven by tissue/cell type-specific and functional specializations. As we will discuss, nucleosome remodelers from each subfamily are canonically associated with distinct mechanisms of nucleosome manipulation and action at specific genomic locations, which influences the classes of ncRNAs they may regulate.

While the DExx/HELICc domains are the catalytic center of nucleosome remodelers, accessory protein domains and remodeler subunits serve important roles in nucleosome remodeler function and regulation. Some subunits contain protein domains that regulate remodeler substrate recognition. For example, accessory bromodomains and chromodomains found in both ATPase and non-ATPase subunits recognize and bind acetylated [20] and methylated [21] histone tails, respectively. While subunits with substrate recognition domains mediate complex targeting and nucleosome interactions, other subunits and functional domains regulate ATPase activity. Actin and actin-related protein subunits of switch/sucrose non-fermentable (SWI/SNF) and inositol requiring (INO80) nucleosome remodelers interact with the resident helicase/SANT-associated (HSA) domains of the ATPase subunits to increase their catalytic activity [22,23,24]. ATPase accessory DNA-binding domains such as HAND-SLIDE-SANT (HSS) and the DNA binding domain (DBD) of imitation switch 1 (ISWI) and chromodomain helicase DNA-binding (CHD) family ATPases, respectively, intrinsically limit ATPase activity until they are bound to the linker DNA between nucleosomes [25,26]. While various subunits of remodelers physically interact with ncRNAs [27,28,29], canonical RNA binding domains such as the RNA recognition motif (RRM) [30] or the La module [31] appear to be absent in nucleosome remodelers. Instead, these interactions are facilitated by novel and non-canonical RNA binding domains distributed on nucleosome remodeler subunits.

## 3. Non-Coding RNAs

Non-coding RNAs are broadly defined as any RNA transcript generated from non-protein coding DNA (meaning any transcript outside of the exons within an mRNA transcript). The first described non-coding RNAs were transfer RNAs (tRNAs) [32], but with advances in transcriptomic and bioinformatic approaches, the numbers and classes of ncRNAs have exploded in recent years [7,33]. Nomenclature and classification of ncRNAs is constantly evolving as the field develops, but current classifications are based upon length, functionality, and/or the genomic region from which the transcript of interest arises. ncRNAs smaller than 200 nucleotides (nt) in length are categorized as small ncRNAs (sncRNAs) while classes >200 nt are categorized as long non-coding RNAs (lncRNAs). Functionally defined sncRNA classes include tRNAs, rRNAs, miRNAs, small nucleolar RNAs (snoRNAs), Piwi-interacting RNAs (piRNAs), and small nuclear RNAs (snRNAs). Classification of lncRNAs arising from an annotated genomic region are often defined by the identity of the associated genomic element. Examples of this include lncRNA classes such as promoter proximal transcripts (PROMPTs), enhancer-derived RNAs (eRNAs), telomeric-repeat containing RNAs (TERRAs), intronic RNAs, and, if no annotated genomic element is associated, intergenic RNAs.

As it has become apparent in recent years that ncRNAs comprise the majority of the eukaryotic transcriptome [34,35,36], research has focused on elucidating the potential functions of these transcripts. miRNAs (reviewed in [37,38]) have an established role in post-transcriptional gene regulation, with well-defined mechanisms through which they regulate the stability of target mRNAs. Generally, miRNAs are first transcribed as long, microRNA-containing non-coding RNAs referred to as primary miRNAs [37,38]. The primary miRNA is then cleaved into 70–120 nt precursor-miRNAs (pre-miRNAs) by the microprocessor complex and exported to the cytoplasm by the nuclear transport protein Exportin-5 [37,38]. In the cytoplasm, the pre-miRNA is further processed by the endoribonuclease Dicer-III and associated RNA binding proteins into a mature single-stranded miRNA (~22 nt in length) [37,38]. The Dicer-III and mature miRNA duplex are incorporated into the RNA-induced silencing complex (RISC), which is guided by the miRNA to the 3’UTR or 5’UTR of target mRNAs due to sequence complementarity to cause translational repression [37,38]. Based upon sequence conservation analysis within 3’UTRs, it is estimated that more than 60% of human protein coding genes are regulated by miRNAs [39]. Given this, it may come as no surprise that the subunits of nucleosome remodeling complexes are regulated by miRNAs, as detailed in the following sections.

Transcripts categorized as lncRNAs are the most numerous [35,40] and functionally diverse category of ncRNAs. Biochemically, lncRNAs are defined as any non-coding RNA greater than 200 nt in length. Given this criteria, 200+ nt non-coding transcripts generated from exons, introns, telomeres, intergenic regions, promoters, or enhancers are viewed as classes of lncRNAs, or ncRNA classes that fall within the category of lncRNAs. The precise number of lncRNAs generated from the human genome is unknown, and is surely cell type dependent, but current estimates place it in the range of tens of thousands to over one hundred thousand transcripts [7,40]. While the majority of these transcripts have not been investigated, studies performed thus far have revealed that lncRNAs regulate gene expression at both the transcriptional and translational level through four major functional models (reviewed in [41,42]). First, lncRNAs can act as a molecular “signal”, with their production signaling the occurrence of a significant biological event. DNA damage-induced lncRNAs may function in this role, but the resulting transcript can have downstream functionalities related to the other functional models. lncRNAs can behave as a “decoy” by interacting with proteins or other RNA species to stop interactions with a primary target (such as genomic DNA or another RNA molecule). lncRNAs functioning as decoys are exemplified in miRNA sinks, which prevent miRNA-mediated silencing of a target mRNA by sequestering miRNAs in place of the target due to sequencing complementarity [43]. The two final functional models are lncRNAs serving as “guides” and “scaffolds”. Guide lncRNAs facilitate the recruitment of protein complexes to genomic loci, while scaffolds serve as the nucleation point for the formation of protein complexes or mediate interactions between different protein complexes. 

These modes of lncRNA functionality are represented with the interactions of the polycomb repressive complex 2 (PRC2) and lncRNAs such as *XIST* [44] and HOTAIR [45]. The PRC2 complex catalyzes the tri-methylation of the lysine at residue 23 of histone H3 (H3K27me3), which is associated with the transcriptional repression of chromatin regions that host the modification. Many copies of *XIST* are transcribed from the copy of the X chromosome within female somatic cells destined to become inactive (Xi), which then forms stable trans-associations with chromatin across Xi. The PRC2 complex forms a physical interaction with *XIST*, which is required for the proper recruitment of PRC2 and deposition of H3K27me3 for initiation of X chromosome inactivation [44]. Thus, *Xist* operates as a guide lncRNA to recruit PRC2 to Xi. The 5’ end of HOTAIR interacts with PRC2 to induce the transcriptional repression of the *HOXD* locus through a similar mechanism to *Xist* [45]. However, HOTAIR also interacts with another chromatin-modifying complex, LSD1/REST/CoREST, to drive the demethylation of lysine residue 4 on histone H3 to facilitate transcriptional repression [46]. Here, HOTAIR acts as a guide between these two protein complexes to alter the epigenomic landscape around target locations. A recent analysis showed that at least 100 human lncRNAs form physical interactions with PRC2 individually, and several others interact with additional chromatin modifying complexes [47]. Studies such as these hint that functional interactions between chromatin modifying complexes and RNAs are more abundant in the nucleus than previously thought and share commonalities with the observed interactions between ncRNAs and nucleosome remodelers.

Currently, non-coding transcription around genomic regulatory elements termed enhancers is a focus point within the field. Enhancers are gene-distal loci capable of regulating the transcription of target genes by increasing the recruitment of RNA polymerase II to target genes [48,49,50]. Enhancers are often (although not always) bi-directionally transcribed into 200+ nt lncRNA transcripts referred to as enhancer RNAs (eRNAs). As enhancers have well-established roles in gene regulation, ongoing research is focused on the role of the ncRNAs at these loci. Genomic and biochemical characterizations of eRNAs have revealed their production correlates well with enhancer activity and they are rapidly degraded by the nuclear RNA exosome [51,52]. Functional characterizations have portrayed a complicated and often conflicting picture, where a handful of eRNAs have important functions in the activity of their associated enhancers while others appear dispensable [53,54]. Identified eRNA functions are centered around physical interactions with chromatin modifying proteins and transcription factors. These interactions mediate the formation and maintenance of chromatin structures critical to enhancer function termed enhancer-promoter loops [53,55,56], extend transcription factor occupancy at enhancers [57,58], or influence RNA polymerase II pause dynamics at target promoters [54,59]. These reported interactions are reminiscent of those described between nuclear proteins and other lncRNA classes, suggesting that this may be a more general functionality for lncRNAs. 

## 4. Non-Coding RNAs Regulate Nucleosome Remodelers through Complex Targeting or Composition

### 4.1. ncRNAs Can Regulate Nucleosome Remodeler Targeting

Similar to histone modifying complexes such as PRC2, the action of nucleosome remodelers can be guided by interactions with ncRNAs (Figure 1). These functional interactions occur in a post-translational manner through diverse mechanisms and have mostly been documented in mammalian systems.

A well-characterized example of this type of interaction is between that of the BAF complex and several lncRNAs. The BAF (BRM/BRG1-associated factor) remodeling complex, the mammalian homolog of the yeast SWI/SNF and fly BAP complexes, consists of approximately 15 subunits filled by several gene families. BAF complex subunit composition can vary with respect to cell-type, which, when considered with the high number of potential BAF subunits, leads to the possibility that hundreds of cell-type specific BAF complexes exist. The BAF complex localizes to gene regulatory sites, such as enhancers and promoters, where it translocates DNA to alter nucleosome positioning. This activity regulates the accessibility of these locations for the core transcriptional machinery and is critical to appropriate gene expression in a variety of cellular processes, including maintenance of pluripotency, neural, heart, and muscle development, and tumor suppression (reviewed in [60]). Targeting of BAF to genomic sites is mediated, in part, through interactions between its non-ATPase subunits and transcription factors, histone modifications, and most recently described, ncRNAs. Direct interactions between ncRNAs and non-ATPase subunits are proposed to modulate BAF complex targeting. A described example of this exists between the lncRNA *SChLAP1* and the BAF subunit SMARCB1/SNF5. SNF5, a core subunit of the BAF complex, is required for proper complex assembly and function [61]. *SChLAP1* was identified as an aberrantly expressed lncRNA in a screen of lncRNAs in prostate cancer tissues [62]. Subsequently, *SChLAP1* was found to directly interact with SNF5 in human prostate cells and overexpression of *SChLAP1* led to a decrease in BAF occupancy genome-wide [63]. In this case, *SChLAP1* negatively influenced BAF occupancy, potentially through acting as a lncRNA decoy to inhibit complex formation through sequestration of SNF5. 

In mammalian cells, the BAF complex can utilize one of two ATPases, Brahma (BRM) [64] or Brahma-related gene 1 (Brg1) [65], which perform the same catalytic activity (DNA translocation) but result in differential gene regulatory outcomes and deletion/depletion of each one lead to different phenotypes [66]. Furthermore, some cell types express higher levels of one ATPase—for example in murine embryonic stem cells, Brg1 is the only ATPase expressed. Brg1 reportedly interacts with several ncRNAs. The lncRNA *Xist*, discussed previously, physically associates with Brg1 and inhibits its ATPase activity in vitro [67]. *Xist* overexpression in an inducible murine embryonic fibroblast system results in decreased Brg1 occupancy across the Xi, arguing that an important function of *Xist* during Xi inactivation is to inhibit BAF targeting to the Xi through decoy interactions with Brg1 [67]. Another lncRNA, *UCA1*, similarly behaves as a decoy to restrict BAF chromatin localization through Brg1 association [68]. As *UCA1* is overexpressed in bladder cancer tissue, this interaction may result in an inhibition of BAF’s tumor suppressor activities and cancer cell proliferation [68]. *lncTC7*, a lncRNA highly expressed in cancer stem cells, binds to the BAF subunits Brg1, BAF170, and SNF5 [69]. In an alternative mechanism relative to *UCA1* or *Xist*, *lncTC7* functions as a guide lncRNA to promote BAF recruitment to the *TCF7* locus. Recruitment of BAF to the *TCF7* gene by *lncTC7* is proposed to drive Wnt signaling and promote cancer stem cell self-renewal [69]. Similar mechanisms of guide lncRNA-mediated BAF targeting occur with inflammation-associated lncRNAs such as *lincRNA-Cox2* [70], *IL-7-AS* [71], and *MALAT1* [72], each resulting in the upregulation of distinct groups of pro-inflammatory response genes. 

Similar interactions between lncRNAs and nucleosome remodeling complex subunits have been reported with other nucleosome remodeling complexes, including the Snf2-related CREBP activator protein (SRCAP) complex, the nucleosome remodeling and deacylation (NuRD) complex, and alpha-thalassemia X-linked mental retardation (ATRX). The SRCAP remodeling complex from the INO80 family regulates chromatin structure by altering the composition of nucleosomes. SRCAP facilitates the exchange of canonical histone H2A with the histone variant H2AZ [73], which itself is a critical regulator of transcription, contributing to important cellular processes. A highly expressed lncRNA in murine embryonic stem cells (mESCs), *lncKdm2b*, interacts with the ATPase of the SRCAP complex, SRCAP [74]. Association with *lncKdm2b* increases SRCAP ATPase activity in mESC nuclear lysates and promotes complex integrity [74]. This interaction is proposed to facilitate SRCAP activity to drive expression of *Zbtb3*, a transcriptional activator that upregulates Nanog expression, therefore maintaining mESC pluripotency [74]. Targeting of another nucleosome remodeler, the NuRD complex, can involve lncRNA interaction as well. Similar to BAF composition, the ATPase of the NuRD complex can be one of two related ATPases: CHD3 or CHD4 of the CHD family [75]. While ATPase identity influences complex targeting, the NuRD complex generally operates as a transcriptional repressor by coupling nucleosome sliding with histone deacetylase activity (the later performed by histone deacetylase, or HDAC, enzymes) [76]. Upon hypotonic stress induction, the NuRD complex is recruited to rRNA genes through an interaction with CHD4 and the lncRNA *PAPAS* [77]. *PAPAS*-dependent NuRD recruitment results in the loss of histone H4 acetylation and repositioning of nucleosomes at the rDNA promoter to downregulate transcription of rDNA. A final example involves the interaction with *Xist* and a less well-characterized nucleosome remodeler, ATRX. ATRX localizes to heterochromatic genomic regions via RNA interactions to modulate DNA methylation and/or nucleosome composition and exchanges histone H3 for the histone variant H3.3 [78,79]. ATRX interacts with *Xist* through a non-canonical RNA-binding domain, and this interaction is required for proper targeting of PRC2 to a subset of Polycomb target genes on Xi [79]. While examples of interactions between nucleosome remodelers and ncRNAs outside of BAF are less frequent at this time, this likely reflects the abundance for BAF-focused research in the field rather than BAF more often interacting with ncRNAs relative to other nucleosome remodelers. Regardless, with the increasing number of reported interactions between nucleosome remodelers and ncRNAs, the intriguing question arises of how these interactions are arbitrated by both partners.

What characteristics determine if or how ncRNAs can interact with nucleosome remodeler subunits? The majority of studies that have identified interactions between ncRNAs and nucleosome remodeler have not probed deep enough to answer these questions. However, a small number of studies have characterized the specific interacting regions between ncRNAs and BAF and ATRX. 7SK snRNA is a central member of the 7SK small nuclear ribonucleoprotein complex, a complex with critical roles in transcriptional regulation [80,81]. Utilizing nucleotide interaction mapping and RNA secondary structure predicting software, Flynn et al. identified a 30 nt stretch of 7SK that interacts with BAF, with a predicted secondary stem loop structure that could mediate this interaction [15]. In a similar manner, a stretch of 200 nt at the 3’ end of *lncTCF7* is necessary to bind BAF subunits BAF170, Brg1, and SNF5, with a stable stem-loop secondary structure predicted to mediate the interaction [69]. Secondary structural studies focused on *Xist* also suggest an important relationship with RNA secondary structure and protein partner interaction. *Xist* holds six repeating sequence elements (referred to as elements A–F) that mediate interactions with its binding partners. ATRX binds *Xist* repeat A with its N-terminal RNA binding domain (RBD) and its C-terminal helicase domain [79]. In comparison, Brg1 interacts with *Xist* repeats A, B, E, and F, though the domains or residues of Brg1 that mediate this interaction have not been determined [67]. Two separate structural studies have shown that the repeat elements of *Xist* are highly structured in vivo, forming secondary structures such as stem-loops [82,83]. When considered together, these studies suggest that secondary structure of ncRNAs likely play a strong role in their interactions with nucleosome remodelers.

If canonical RNA binding domains are absent in remodeler subunits, what protein domains could facilitate their interactions with ncRNAs? In 2018, Rahnamoun et al. examined how the enhancer localization of the chromatin associated protein BRD4 was influenced by eRNAs produced at these locations in human colon cancer cells [57]. BRD4 is a transcriptional regulator that binds acetylated histone tails through its two tandem bromodomains [57]. Rahnamoun et al. found that the bromodomains of BRD4 and several other bromodomain-containing proteins, including Brg1, were capable of binding eRNAs generated from the *MM9* and *CCL2* enhancers [57]. Thus, Brg1-ncRNA interactions can be mediated through its N-terminal bromodomains. It remains unclear how these bromodomain interactions might regulate BAF function, but as interactions with eRNAs increased BRD4 occupancy at enhancers, this opens the possibility of a similar mechanism between eRNAs and Brg1. Cajigas et al. found that the lncRNA *Evf* binds Brg1 within its HSA and BRK domains to downregulate Brg1 ATPase activity [84]. Functionally, these domains likely mediate protein–protein interactions [22,85], and therefore this outcome could be the result of *Evf* behaving as a decoy to influence BAF complex stability or extra-complex interactions associated with targeting. A third study, by Han et al. found that the helicase domain of Brg1 binds the lncRNA *Mhrt779* with high affinity in vitro [86]. This interaction reduces Brg1 occupancy at the promoter of three BAF-regulated genes important in cardiac stress response, *Myh6*, *Myh7*, and *OPN* in rat primary cardiomyocytes [86]. Together, these studies illustrate how Brg1 can interact with ncRNAs through several protein domains. It should be noted that Brg1 could interact with different eRNAs through its bromodomains [57] and several lncRNAs and RNA competitors with its HSA/BRK domains [84]. Thus, Brg1-RNA interactions may be indiscriminate to transcript sequence, and instead rely upon other factors such as proximity, length, or secondary structure; it remains to be determined if the highly conserved helicase domain of Brg1 follows this trend. But what of nucleosome remodelers beyond BAF and ATRX? A string of recent screens examining the proteins that bind ncRNAs in human cell lines may help to answer this question [28,29,87]. These screens identified ATPases from the ISWI subfamily [29,87], and CHD, INO80, and SWI/SNF subfamilies [28,29,87], and several subunits from INO80 family nucleosome remodelers [28,29,87] as interactors of non-poly-adenylated ncRNAs and nascent transcripts. While many of these interactions await validation and the participating ncRNAs remain unidentified, these screens support that associations between ncRNAs and nucleosome remodelers are a widespread phenomenon in mammalian systems. Even still, while several BAF complex members can bind 7SK [15] and *lncTCF7* [69], it remains unclear which protein domain(s) of each subunit mediate these interactions. Identifying which remodeler domains or residues are capable of binding RNAs and the specificity of these interactions are critical questions to answer.

### 4.2. ncRNAs Can Alter Nucleosome Remodeler Complex Composition

ncRNAs can regulate nucleosome remodeler activity at the post-transcriptional level (Figure 1). While the majority of described mechanisms are carried out by miRNAs, examples of lncRNA mediated mechanisms have also been described. 

miRNA-based regulation of nucleosome remodelers can result in two outcomes: fine tuning the expression of individual subunits, and mediating subunit composition. The expression level of several subunits from SWI/SNF [88,89], ISWI [90], CHD [91,92], and INO80 [93,94] nucleosome remodeler families are regulated by miRNAs. Many of these interactions were initially identified through over-expression of characterized miRNAs and/or specific remodeler subunits in cancer types, or through correlated expression profiles with identified miRNAs [88,89,90,91,92,93,94]. For example, BAF subunit BAF60A expression is mis-regulated in two cancer types [88,89]. In two cases, transcriptomic analyses identified a differentially-expressed miRNA (miR-7 [88] and miR-490-3p [89]) that was predicted to target BAF60A via an in silico approach. These interactions were later validated through complementary in vitro and in vivo experiments. These miRNAs target the 3’UTR of *SMARCD1* (which encodes BAF60A) to downregulate its expression post-transcriptionally [88,89]. In normal conditions, regulation by these two miRNAs likely fine tunes the expression of BAF60A to regulate BAF activity, but when the expression of either miRNA becomes aberrant as in the described cancers, this regulation likely impacts important BAF functions, such as its known tumor suppressor activity. Relatedly, CHD1 was recently identified as a target for three different liver-specific miRNAs associated with dietary intake in mice [91]. This study identified that these miRNAs decrease the protein abundance of CHD1 to varying degrees (between 20–40%). This observation illustrates a key point of miRNA regulation: the use of alternative miRNAs can fine tune target protein abundance to balance levels without reducing overall abundance completely. Given that miRNAs are predicted to regulate at least half of human genes (reviewed in [95]), it is likely that undescribed miRNAs fine tune nucleosome remodeler activity through targeting subunits beyond those described to date. 

By reducing the abundance of nucleosome remodeler subunits, miRNAs can stimulate the exchange of subunits and alter complex composition. A well-characterized example of this phenomenon is the altering subunit composition of the BAF complex during mammalian neural development. In neural stem cells, the BAF complex exists as a cell-type specific assembly termed neural progenitor BAF (npBAF). During differentiation from neural stem cells to post-mitotic neural cells, npBAF undergoes a series of subunit exchanges to become a neural cell specific assembly referred to as neural BAF (nBAF). Among these exchanges is the swapping of BAF53a with BAF53b. BAF53a and BAF53b belong to the same gene family and are critical for BAF ATPase activity, but do not exist within the same complex [96]. This BAF subunit exchange is regulated through miRNA-mediated down regulation of BAF53a during differentiation, by two miRNAs, miRNA9* and miRNA124 [97]. Prior to differentiation, these two miRNAs are repressed by the transcriptional repressor REST. However, REST activity is repressed by the unliganded retinoic acid receptor (RAR) complex at the onset of differentiation, which permits for activation of miRNA9* and miRNA124 expression. miRNA9* and miRNA124 repress the abundance of BAF53a, which drives the replacement of BAF53a with BAF53b for nBAF assembly [97]. Additional unique subunit exchange of BAF components during cell differentiation are known, and, while currently unknown, miRNA-mediated regulation of BAF composition may be more pervasive, and this may perhaps extend to other nucleosome remodelers with cell specific activity. 

Cytoplasmic lncRNAs are primarily involved in post-transcriptional regulation of target mRNAs. The lncRNA *CR993309* has been implicated as a post-transcriptional regulator of the INO80 complex subunit D (INO80D) in human epithelial cells [94]. This interaction was initially identified through a genome wide in silico approach looking for annotated lncRNAs with sequence complementarity to the 3’UTR of coding genes. *CR993309*, overlapping the *INO80D* gene, bears 99% sequence homology to an approximately 9kb region of the 3’UTR of *INO80D* [94]. miRNA-5096 targets the 3’UTR of *INO80D* [94] and results in reduced expression of *INO80D*. Cells overexpressing *CR993309* transfected with miRNA-5096 show an increase in *INO80D* expression, suggesting that *CRR993309* may act as a ‘miRNA sponge’, or a decoy that sequesters miRNA-5096 to interfere with suppression of its intended target, *INO80D* [94]. Other examples of lncRNAs acting as miRNA decoys are prevalent (reviewed in [98]), and therefore screens for lncRNAs with sequence complementarity to the 3’UTRs of nucleosome remodeler subunits may reveal novel examples of this type of ncRNA-based regulation.

## 5. Members from Each Nucleosome Remodeling Family Regulate ncRNAs

In comparison to the numerous mechanisms by which ncRNAs modulate nucleosome remodeler function, the mechanisms by which nucleosome remodelers regulate ncRNAs are currently limited. ncRNA regulation can occur pre-transcriptionally through limiting the transcription of non-coding loci or through co-/post-transcriptional degradation of the nascent transcript; nucleosome remodelers from each major subfamily have emerged as significant players in the pre-transcriptional regulation of ncRNAs. However, as the number and biological significance of ncRNAs continues to grow, so too does the need to identify mechanisms of ncRNA regulation. In this light, nucleosome remodelers represent a logical first avenue for exploratory research defining novel mechanisms of ncRNA regulation for two significant reasons. First, nucleosome remodelers alter chromatin structure through diverse ATP-dependent mechanisms (i.e., nucleosome positioning, occupancy, and editing), and some nucleosome remodelers demonstrate the capacity to utilize more than one of these mechanisms in different context [99,100,101,102]. Second, nucleosome remodelers from each subfamily are only targeted to specific genomic loci, and therefore are likely limited and specific in the ncRNA species they can pre-transcriptionally regulate. When consider together, the potential pathways through which nucleosome remodelers can regulate ncRNAs appears much broader than the field currently understands.

### 5.1. SWI/SNF Family Remodelers Regulate ncRNA Expression at Gene Regulatory Sites in Several Eukaryotic Systems

The relationship between the mammalian SWI/SNF or BAF complex and ncRNAs at gene regulatory sites has been well studied in eukaryotic systems (Figure 2a–c). Universally, SWI/SNF family members mobilize or eject nucleosomes to regulate chromatin accessibility at the regions to which they are recruited, and thus have been found to both promote and repress non-coding transcription in several different systems. The mammalian embryonic stem cell-specific BAF assembly, termed esBAF, localizes to gene regulatory elements within mESCs to modulate nucleosome occupancy [103,104,105]. At these locations, esBAF reinforces the occupancy of enhancer or promoter flanking nucleosomes to suppress eRNA and PROMPT expression, respectively [103]. Further, esBAF was found to suppress non-coding transcription at the 3’ transcription termination sites (TTS) of target genes [103]. This TTS-associated ncRNA regulatory function is evolutionarily conserved with SWI/SNF homologs in *Arabidopsis thaliana* [106]. BAF function is critical for enhancer activation, which promotes eRNA production, in differentiating murine mesoderm progenitor cells [107]. In *Danio rerio*, BAF is required for myogenic miRNA expression in skeletal muscle development [108]. In *S. cerevisiae*, which lack distal enhancer elements, SWI/SNF family nucleosome remodelers RSC and yeast SWI/SNF localize to promoters to control gene activation through chromatin accessibility [109,110,111]. Interestingly, while both RSC and yeast SWI/SNF activity at promoter elements is associated with increased divergent, non-coding transcription [110,111], RSC also suppresses the expression of a subset of divergent lncRNAs analogous to SWI/SNF homologs in higher eukaryotes [112]. The selective, bi-modal nature of ncRNA regulation at gene regulatory sites by SWI/SNF nucleosome remodelers is a dynamic process, most likely driven by site-specific cues that are constantly fluctuating in response to the changing cellular state.

### 5.2. INO80 Family Nucleosome Remodelers Likely Influence the Expression of Many ncRNA Species

INO80 family nucleosome remodelers, divided into the SWR1 and INO80 sub-classes, alter chromatin structure primarily through regulating the chromatin localization of histone variant H2AZ (Figure 2d,e). Histone variant H2AZ (a variant of canonical histone H2A) is an evolutionarily conversed histone variant that plays a critical role in many cellular processes through transcriptional regulation from yeast to human (reviewed in [113]). SWR1 sub-class nucleosome remodelers such as yeast SWR1, along with related homologs in higher eukaryotes, have an evolutionarily conserved role in H2AZ deposition onto chromatin [73,114,115]. In yeast, SWR1 catalyzes the deposition of H2AZ at target promoters to facilitate their transcriptional activation [114]. SWR1 sub-class nucleosome remodelers SRCAP and p400 (part of the Tip60-p400 complex) facilitate the deposition of H2AZ into nucleosomes flanking enhancers and promoters in mice [73,115]. At enhancers, deposition of H2AZ is associated with upregulated enhancer activity and increased eRNA production [116], making SWR sub-class remodelers such as SRCAP and Tip60-p400 complexes likely activators of eRNA expression in mammals. Interestingly, in *Arabidopsis thaliana*, the SWR1 homolog SWR1-C is required for proper transcriptional activation of miRNAs critical to flower and leaf development [117], suggesting roles for ncRNA activation for SWR-subclass INO80 nucleosome remodelers beyond regulatory elements. In higher eukaryotes, H2AZ removal from chromatin is catalyzed by INO80 and related homologs, while this function is disputed for INO80 in *S. cerevisiae* [118,119,120,121]. Loss-of-function assays targeting the ATPase of INO80 overlaps with the aberrant accumulation of H2AZ genome wide in yeast and mESCs [120,122]. This coincides with pervasive, cryptic transcription within the bodies of ~43% of yeast genes and telomeres of chromosomes in yeast [120,123], as well as increased PROMPT and cryptic intergenic ncRNA expression at a subset of target genes in mESCs [122]. These studies suggest a conserved role for INO80 in ncRNA suppression genome wide, influencing several ncRNA species, which may be related to the histone variant H2AZ.

### 5.3. ISWI Nucleosome Remodelers Are Critical Regulators of Intragenic Cryptic ncRNAs.

ISWI nucleosome remodelers are typically associated with nucleosome remodeling activity that promotes chromatin organization, utilizing nucleosome sliding to establish regularly spaced nucleosome arrays following chromatin altering events such as replication or transcription [124]. Localization and transcriptomic analyses in yeast have shown that after the replicational or transcriptional machinery passes through the gene body, ISWI nucleosome remodelers reposition nucleosomes into an evenly spaced array that promote chromatin compaction and suppress cryptic initiation of ncRNAs within gene bodies [125,126,127,128]. Cryptic, intragenic transcription interferes with the transcription of coding genes [112]; thus, this suppressive regulatory role by ISWI likely promotes genomic stability (Figure 2f). Interestingly, suppression of divergent transcription at promoters in yeast is maintained in part through the remodeling activity of ISWI nucleosome remodelers [126] (Figure 2b). In direct opposition to yeast SWI/SNF and RSC nucleosome remodeler activity, yeast ISWI nucleosome remodelers decrease the distance between flanking nucleosomes of the promoter nucleosome depleted regions (NDRs) of target genes. This activity likely limits accessibility to transcription start sites within target promoters, as it is associated with suppressing both coding and non-coding transcription [126]. The members and functions of ISWI family nucleosome remodelers are diversified in higher eukaryotes, but studies in *Drosophila melanogaster* [129,130], human cell lines [131,132,133], and murine epithelial cells [134] suggest these mechanisms of nucleosome array establishment and restriction of NDR size are evolutionarily. Given this, the suppression of cryptic transcription within gene bodies and heterochromatic regions (Figure 2g) and divergent, non-coding transcription at promoters by ISWI nucleosome remodelers is likely a common trait of eukaryotes.

### 5.4. CHD Nucleosome Remodelers Show Significant Potential as ncRNA Regulators 

CHD nucleosome remodelers are an extraordinary example of nucleosome remodeler diversification in higher eukaryotes. While *S. cerevisiae* and *Schizosaccharomyces pombe* have one and four representative members of this family, respectively, mammalian cells have at least 9 CHD family nucleosome remodelers (reviewed in [135]). In higher eukaryotes, CHD remodelers are distributed among three CHD subclasses (CHD1-2, CHD3-4, and CHD5-9) which are differentiated by the presence of additional functional domains within the ATPase subunit [135]. Like yeast ISWI nucleosome remodelers, yeast CHD1 arranges nucleosomes into evenly spaced arrays in the wake of the transcriptional machinery, which suppresses expression of intragenic cryptic ncRNAs [124,128,130]. While CHD1 homologs in higher eukaryotes modulate nucleosome positioning similar to yeast CHD1, other CHD nucleosome remodelers have been implicated in regulating chromatin structure through both nucleosome eviction and editing [101]. As discussed, mammalian CHD4 regulates rRNA expression through modulating nucleosome positioning [77]. However, fly CHD1 is critical for the proper deposition of histone variant H3.3 in developing fly male gametes, demonstrating a role for CHD nucleosome remodelers in nucleosome editing. A recent study of *S. pombe* found that depletion of two CHD nucleosome remodelers, Hpr1 and Hpr3 (homologs of yeast/human CHD1), leads to an increase in nucleosome occupancy at promoters genome-wide, which suggests CHD remodelers can modulate nucleosome occupancy as well [136]. Several CHD nucleosome remodelers, including CHD7 [137] and CHD8 [138], are known to localize to enhancer elements in human cell lines. While the functions of these factors at enhancers are unknown, the localization of these nucleosome remodelers to each target location is imperative for proper enhancer-driven transcriptional activation of nearby genes [137,138]. Currently, CHD1 and CHD4 are the only known CHD nucleosome remodelers with established ncRNA regulatory roles. However, given the broad diversity of CHD nucleosome remodelers in higher eukaryotes and their diverse mechanisms of action, it is likely that novel mechanisms of ncRNA regulation by CHD nucleosome remodelers await characterization.

## 6. Conclusions and Future Perspectives

Nucleosome remodelers and ncRNAs regulate one another through a complicated, multi-faceted, and partially understood interactions. On one side of this interaction are nucleosome remodelers, whose activity is required for ncRNA expression in some instances but necessary for their suppression to maintain genomic stability in others. On the other side of this interaction are ncRNAs, indispensable to proper nucleosome remodeling targeting and complex composition yet serving as critical limiters for the functions of others. Multiple ncRNAs in seemingly redundant fashion exist to keep in check the abundance of one nucleosome remodeler. In a more complicated fashion, ncRNAs exist to regulate ncRNAs that regulate nucleosome remodelers. In terms of mechanistic diversity, ncRNAs boast an interesting array of methods to regulate nucleosome remodeler function at many levels. While the mechanisms are diverse, individual nucleosome remodeler regulation by specific ncRNA(s) is highly explicit, targeting distinct remodelers or influencing only particular types of genomic elements or regions. In this regard, the popular notion of ncRNAs functioning as a ‘fine tuner’ of cellular processes is an attractive, and inclusive, model. With the high number of ncRNAs awaiting characterization, it will be exciting to see the scope of current mechanisms of nucleosome remodeler regulation and witness the discovery of novel ones. In comparison, ATP-dependent activities by nucleosome remodelers regulate ncRNAs primarily at the transcriptional level. Still, multiple members from each nucleosome remodeler subfamily in evolutionarily diverged systems modulate ncRNA expression through discrete, nucleosome-based mechanisms. This highlights that remodelers have an evolutionarily conserved role in ncRNA regulation that has only grown over evolutionary time, likely due to increasing eukaryotic genome size and complexity.

Even with the substantial progress that has been made, important questions and limitations in our understanding remain about the regulatory interactions between ncRNAs and nucleosome remodelers. The functions for many ncRNAs await characterization, but current approaches to elucidate RNA functions are labor intensive and low throughput. The development of new techniques and approaches that can elucidate ncRNA functions in a systematic and genome-wide scale will expediate this research. Until these techniques are developed and applied, the full scope of remodeler subunits regulated by miRNAs and lncRNAs will remain unresolved. Another important question to address is how RNA secondary structure mediates ncRNA interactions with nucleosome remodelers. A handful of studies suggest that secondary structure plays a more influential role in mediating ncRNA-nucleosome remodeler interactions than transcript length or sequence, but only additional secondary structural studies of ncRNA regions that come in physical contact with nucleosome remodeler subunits will address this issue. In addition, it will be important to identify and validate ncRNA binding interactions for novel nucleosome remodeler subunits, as well as identify the RNA binding protein domains and the functional implications of these binding events on remodeling activity. These data will clarify the commonality of ncRNA interactions with nucleosome remodelers and inform our perspective of its impact on remodeler function. While outside the scope of this review, understanding how non-ATP-dependent mechanisms of ncRNA regulation fit into the network of ncRNA and nucleosome remodeler interactions will undoubtedly be important. Activities such as histone post-translational modifications and extra-complex protein interactions alter chromatin structure and transcription in alternative ways and likely shapes the non-coding transcriptome through new mechanisms.

The current body of research has shown that ncRNAs and nucleosome remodelers regulate one another through a complex, yet balanced, network of interactions. Concurrently, research has illuminated the depth of what remains to be determined regarding this network. With the vast amount of knowledge left to be unlocked in this field, it is an exciting time for research in chromatin and ncRNA biology.

## Figures and Tables

**Figure 1 biology-09-00213-f001:**
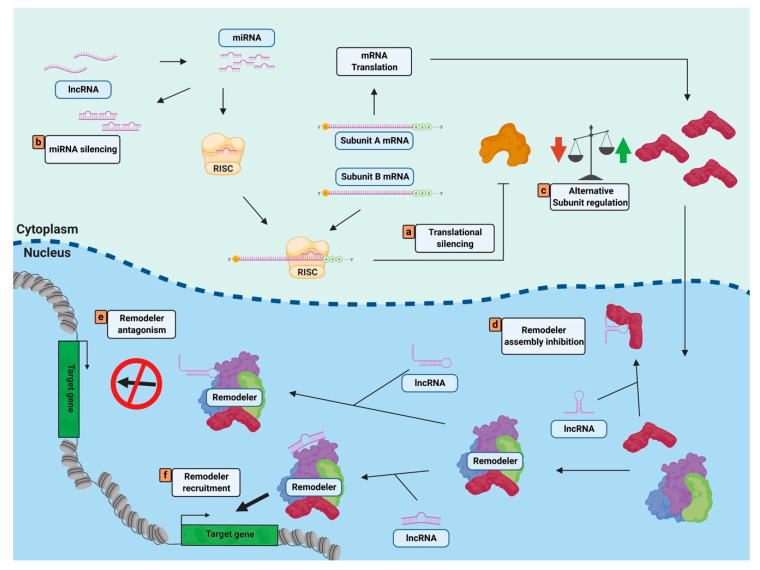
Examples of non-coding RNAs regulating nucleosome remodelers. (a–f) ncRNAs of various classes can regulate the activity of nucleosome remodelers pre-transcriptionally through (a) translational silencing, (b) miRNA silencing, or (c) alternative subunit regulation and post-transcriptionally through (d) remodeler assembly inhibition, (e) remodeler antagonism, or (f) remodeler recruitment. Figure created with Biorender.com.

**Figure 2 biology-09-00213-f002:**
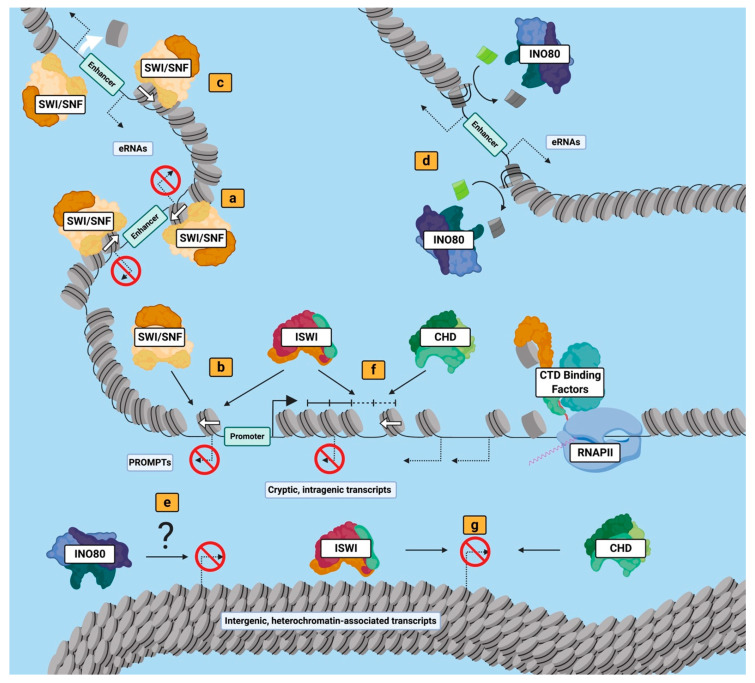
Mechanisms of ncRNA regulation by nucleosome remodelers of the four major subfamilies in higher eukaryotes. (a–c) Nucleosome remodelers from the SWI/SNF family utilize nucleosome sliding (a) and (b) and nucleosome eviction (c) to promote or suppress expression of ncRNAs associated with regulatory elements (eRNAs and PROMPTs). (d,e) Members of the INO80 family regulate promote PROMPT expression through nucleosome editing (d) but suppress heterochromatin-associated transcripts through less well-understood mechanisms (e). (f–g) Nucleosome remodelers of the ISWI and CHD families modulate nucleosome positioning to establish evenly spaced arrays of nucleosomes, to suppress intergenic (e) and intragenic (g) ncRNA expression. Figure created with Biorender.com.

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
