# Peer review of "Non-Coding RNAs and Nucleosome Remodeling Complexes: An Intricate Regulatory Relationship"

_biology, 2020, doi:10.3390/biology9080213_

Round 1

Reviewer 1 Report

  1. Please rephrase sentence in line 39-40, for better understanding by the reader - “Traditionally, RNA transcripts generated from these regions of non-protein-coding 39 genomic “dark matter” were thought of as random transcriptional noise resulting in so-called “junk”.”
  2. Line 52, “interesting trend in chromatin research” – can it be more specific than trend, why it is important in the field of chromatin, what are the main findings?
  3. The paragraph in lines 50-58 should be after describing to the reader the meaning and importance of remodeler, what is Nucleosome and the role of remodeling it, including main findings and importance.
  4. Line 88-89, “RNA recognition module (RRM)[30] or the La module[31]”- RRM- RNA recognition Motif and not module, please change. Please describe what is La module.
  5. Noncoding RNAs section is professionally written.
  6. Line 180- as it is a review paper and not a research one the tite should be changed from results to other phrase such as main findings.
  7. Line 181- should change to different sentence such as the main mechanisms that Noncoding RNAs regulate nucleosome remodelers, these different mechanisms should be stated and cleared before starting the sections 2.1.1 and 2.1.2 or alternatively discussed in more detailed in the beginning of each section in order to make it clearer to the reader.
  8. Section 2.2 is well described and organized.

Author Response

  1. Please rephrase sentence in line 39-40, for better understanding by the reader - “Traditionally, RNA transcripts generated from these regions of non-protein-coding 39 genomic “dark matter” were thought of as random transcriptional noise resulting in so-called “junk”.”

Thank you for the suggestion. We have modified the text to read more clearly: Traditionally, RNA transcripts generated from these regions of non-protein-coding genomic regions were described as random transcriptional noise resulting in so-called “junk”. (lines 39-40)

  1. Line 52, “interesting trend in chromatin research” – can it be more specific than trend, why it is important in the field of chromatin, what are the main findings?

Importantly, this is the focus of the entire review, and therefore we cannot expand easily in one sentence. We have altered the sentence to help remove any issues the reviewer had (line 51-52).

  1. The paragraph in lines 50-58 should be after describing to the reader the meaning and importance of remodeler, what is Nucleosome and the role of remodeling it, including main findings and importance.

Rather than following this suggestion, we have done as reviewer 2 suggested for the structure. 

  1. Line 88-89, “RNA recognition module (RRM)[30] or the La module[31]”- RRM- RNA recognition Motif and not module, please change. Please describe what is La module.

We have corrected to replace module with motif. The La module is found in LARP proteins, composed of La motifs (LaM) and RRMs (https://www.ncbi.nlm.nih.gov/pmc/articles/PMC5647580/)

  1. Noncoding RNAs section is professionally written.

Thank you

  1. Line 180- as it is a review paper and not a research one the tite should be changed from results to other phrase such as main findings.

Thank you, we have fixed these headings

  1. Line 181- should change to different sentence such as the main mechanisms that Noncoding RNAs regulate nucleosome remodelers, these different mechanisms should be stated and cleared before starting the sections 2.1.1 and 2.1.2 or alternatively discussed in more detailed in the beginning of each section in order to make it clearer to the reader.

We have modified (now line 184) to be more clear. However, we feel that the review goes into details regarding the mechanisms clearly in the sections and therefore have not added the suggested additional introduction. 

  1. Section 2.2 is well described and organized.

Thank you

Reviewer 2 Report

The presented review by Patty & Hainer discusses the interconnected role of ncRNAs and nucleosome remodelers comprehensively and very detailed. This work represents an important contribution to the field and provides detailed insights into the complex topic of ncRNAs and nucleosome remodeling.

The authors very well structured the content and in my eyes included the most important chapters, therefore, I only have minor comments on the style and structure of the manuscript.

  • The whole manuscript is very well written and the english language used is a pleasure to read (which should be the case for every submitted manuscript, but reality often is different). The authors put lots of efforts into writing style, which makes the reading of the manuscript enjoyable.
  • Please put spaces between the [REF] and the word in front
  • be careful to write the full name of an abbreviation at first use e.g. line 84 HSA, line 86 ISWI, CHD --> please check whole manuscript
  • write in vitro and in vivo cursive (e.g. line 212)
  • maybe shorten reference 8 - e.g. list 20 authors and place et al. after author 20
  • Please modify the structure – as this is a review, I don’t think it is appropriate to put in a „RESULTS“ section. I would suggest to end the introduction after line 58, since the authors give an outlook what the review will discuss, which for me is a perfect ending for an introduction.

    As already mentioned skip the results part and start numbering the sections (2. …) with Nucleosome remodelers (line 60)

    After 2.2.4 follows 3.2 Figures, Tables and Schemes – I think it is confusing to put an extra paragraph with figures. Figures should be mentioned and cited in the text and should be placed near their mentioning within the text!

Overall, I think the review is very well written, gives lots of important information on a current topic and I am sure that the scientific community will definitely benefit from reading it.

Author Response

The presented review by Patty & Hainer discusses the interconnected role of ncRNAs and nucleosome remodelers comprehensively and very detailed. This work represents an important contribution to the field and provides detailed insights into the complex topic of ncRNAs and nucleosome remodeling.

The authors very well structured the content and in my eyes included the most important chapters, therefore, I only have minor comments on the style and structure of the manuscript.

  • The whole manuscript is very well written and the english language used is a pleasure to read (which should be the case for every submitted manuscript, but reality often is different). The authors put lots of efforts into writing style, which makes the reading of the manuscript enjoyable.

Thank you

  • Please put spaces between the [REF] and the word in front

We have fixed this throughout (many locations)

  • be careful to write the full name of an abbreviation at first use e.g. line 84 HSA, line 86 ISWI, CHD --> please check whole manuscript

Thank you, we have corrected these (lines 67, 83, 84, 86)

  • write in vitro and in vivo cursive (e.g. line 212)

We have fixed this throughout (multiple locations, including line 212)

  • maybe shorten reference 8 - e.g. list 20 authors and place et al. after author 20

We have done as suggested (line 574)

  • Please modify the structure – as this is a review, I don’t think it is appropriate to put in a „RESULTS“ section. I would suggest to end the introduction after line 58, since the authors give an outlook what the review will discuss, which for me is a perfect ending for an introduction.

Thank you, we have done as suggested. (multiple headers throughout)

As already mentioned skip the results part and start numbering the sections (2. …) with Nucleosome remodelers (line 60)

Thank you, we have done as suggested, as above.

After 2.2.4 follows 3.2 Figures, Tables and Schemes – I think it is confusing to put an extra paragraph with figures. Figures should be mentioned and cited in the text and should be placed near their mentioning within the text!

Thank you, we have done as suggested (rearranged figures and included citations of Figures in text).

Overall, I think the review is very well written, gives lots of important information on a current topic and I am sure that the scientific community will definitely benefit from reading it.

Thank you!